# Disruption of the Expression of the Placental Clock and Melatonin Genes in Preeclampsia

**DOI:** 10.3390/ijms24032363

**Published:** 2023-01-25

**Authors:** Aïssatou Bailo Diallo, Benjamin Coiffard, Raoul Desbriere, Maria Katsogiannou, Xavier Donato, Florence Bretelle, Soraya Mezouar, Jean-Louis Mege

**Affiliations:** 1MEPHI, IRD, APHM, Aix-Marseille Univ, 13005 Marseille, France; 2Institut Hospitalo-Universitaire Méditerranée-Infection, 13005 Marseille, France; 3Department of Obstetrics and Gynecology, Hôpital Saint Joseph, 13008 Marseille, France; 4Department of Obstetrics and Gynecology, APHM, Hôpital de la Conception, 13005 Marseille, France; 5Department of Immunology, APHM, Hôpital de la Conception, 13005 Marseille, France

**Keywords:** circadian rhythm, placental cells, macrophages, trophoblasts, melatonin

## Abstract

Circadian rhythms have been described in numerous tissues of living organisms and are necessary for homeostasis. The understanding of their role in normal and pathological pregnancy is only just emerging. It has been established that clock genes are expressed in the placenta of animals and humans, but the rhythmicity of placenta immune cells is not known. Macrophages from healthy placenta of women at term were isolated and the expression of clock genes BMAL1, CLOCK, PER2, CRY2, and NR1D1 was assessed by qRT-PCR every 4 h over 24 h. Raw data were treated with cosinor analysis to evaluate the significance of the oscillations. Placental macrophages exhibited significant circadian expression of clock genes but one third of placental macrophages lost clock gene rhythmicity; the clock gene oscillations were restored by co-culture with trophoblasts. We wondered if melatonin, a key hormone regulating circadian rhythm, was involved in the oscillations of placental cells. We showed that macrophages and trophoblasts produced melatonin and expressed MT2 receptor. In women who developed preeclampsia during pregnancy, circadian oscillations of placental macrophages were lost and could not be rescued by coculture with trophoblasts from healthy women. Moreover, production and oscillations of melatonin were altered in preeclamptic macrophages. For the first time to our knowledge, this study shows circadian rhythms and melatonin production by placental macrophages. It also shows that preeclampsia is associated with a disruption of the circadian rhythm of placental cells. These results represent a new scientific breakthrough that may contribute to the prevention and treatment of obstetrical pathologies.

## 1. Introduction

Molecular clocks are internal regulators of biological processes with oscillations that last for approximately 24 h named circadian rhythms [1,2]. In humans, the endogenous biological rhythm is generated in the suprachiasmatic nucleus (SCN) of the hypothalamus [3] through a series of feedback loops coordinated by clock genes. They mainly comprise several transcription factors and include brain and muscle Arnt-like protein (BMAL1, the main gene of the clock regulator), circadian locomotor output cycles kaput (CLOCK), the genes encoding the proteins period (PERs), cryptochromes (CRYs), D-site-binding protein (DBP), REVERBs, and retinoic acid receptor-related orphan receptors (RORs) [4]. The SCN acts as a central pacemaker that synchronizes a lot of peripheral clocks in several tissues [5]. The disruption of the clock genes is associated with alterations of homeostasis [2,6].

A circadian time-keeping system has been observed during pregnancy. Hence, pregnancy parameters including temperature, blood pressure, leukocyte count, weight gain, intra-amniotic fluid pressure, and uterine contractions exhibit circadian oscillations in normal pregnancy [7,8,9]. In preeclampsia, a major complication of pregnancy associated with placenta dysfunction, the disruption of the circadian rhythm of the maternal blood pressure has been reported suggesting that circadian rhythm disorder would be a risk factor for preeclampsia [10]. As obstetrical pathologies such as chorioamnionitis, spontaneous abortion, premature delivery, low birth weight, or preeclampsia are associated with placental dysfunctions [11,12], the circadian rhythmicity of placenta and its alterations is a critical issue.

The placenta is a complex organ found at the interface between the fetus and its maternal host [13]. It plays a key role in nutrition and protection of the fetus by creating an immunological barrier between the maternal circulation and the fetus [14]. Two major cell types are present in the placenta, trophoblast cells, encompassing several subtypes of placental epithelial cells, and immune cells. These latter consist of lymphocytes, natural killers (NK), macrophages, dendritic cells, and mast cells [15]. The NKs are the major population during the first trimester (70%) followed by macrophages that are present during all three trimesters [16,17]. Several clock genes such as BMAL1, CLOCK, CRY1, and PER2 clock genes are expressed by animal and human placenta tissue [18,19,20,21,22] as well as extravillous trophoblast cell lines [23]. However, only the expression of BMAL1 and CLOCK genes exhibit circadian rhythm, suggesting the presence of a peripheral clock in the human placenta [20]. 

During pregnancy, several placental hormones play key roles in maintaining parturiency. Among them, melatonin, a neuroendocrine hormone produced by the pineal gland presents a well-documented role in the regulation of the circadian rhythm [24]. This hormone is synthesized from serotonin by two key enzymes including arylalkylamine N-acetyltransferase (AANAT) and N-acetylserotonin O-methyltransferase (ASMT) and binds two receptor MT1 (Me11a) and MT2 (Me11b) on target cells [25]. Its secretion shows a powerful daily oscillation with only one peak over the 24 h during the night [26] permitting the synchronization of peripheral clocks in mammals [24]. During pregnancy, melatonin is secreted by the placenta [27], but it has been proposed that melatonin secreted by the pineal gland leads to the programming of fetal circadian rhythmicity [28]. There are several pieces of evidence that highlight a key role of placenta melatonin in the regulation of local circadian rhythm: (i) the variation of gene and protein expression of synthesizing-enzymes and receptors of melatonin are reported throughout trimesters [28,29] (ii) trophoblasts express synthesizing-enzymes and receptor of melatonin [29] (iii) melatonin improves morphological differentiation of villous trophoblasts [29].

Here, we report that primary placental macrophages from human full-term placenta exhibited circadian expression of the following clock genes BMAL1, CLOCK, PER2, CRY2, and NR1D1. We observed that the circadian rhythmicity was lost in placental macrophages but restored by culture with trophoblasts. We also found that the circadian rhythm of placental macrophages was disrupted in preeclampsia and was not restored after co-culture with trophoblasts. This finding suggests that the perturbation of the molecular clock in placental macrophages plays a role in preeclampsia. Focusing on melatonin, we report that trophoblasts and macrophages express synthesizing-enzymes and MT2 receptor of melatonin, which was decreased in preeclampsia. Moreover, circadian production of melatonin was altered in the placental macrophage from preeclamptic women. Taken together this study highlights the role of the circadian rhythm in placenta during pregnancy and in obstetrical complications.

## 2. Results

### 2.1. Circadian Expression of Clock Genes in Placental Cells

The expression of the following clock genes *BMAL1*, *CLOCK*, *PER2*, *CRY2*, and *NR1D1*, was measured in macrophages isolated from human full-term placentas every 4 h over 24 h. Using the cosinor model we found two different profiles of expression. In about 70% of them (9/13) the expression of *BMAL1*, *CLOCK*, *CRY2,* and *NR1D1*, but not that of *PER2* gene, exhibited a circadian rhythm with an acrophase occurring at CT16 (Figure 1A and Table 1). In macrophages from about one third of the placentas (4/13), the clock genes did not present a circadian rhythm for all clock genes (Figure 1B). Taken together these results show for the first time the circadian expression of clock genes in placental macrophages.

### 2.2. Trophoblast-Macrophage Cooperation in the Regulation of Placental Circadian Rhythms 

As culture conditions affect circadian oscillations in different cell types including macrophages [30,31], we wondered if placenta macrophages lost the circadian oscillations of the clock genes in culture. We studied the evolution of the circadian rhythm of the clock genes in placental macrophages cultured for four days. The 4-day culture of placental macrophages completely abolished the rhythmicity of *BMAL1*, *CLOCK*, *PER2*, *CRY2,* and *NR1D1* genes (Figure 2A and Table 1).

Then, we incubated placental macrophages cultured for 4 days with high concentrations of serum to synchronize them. The procedure is known to efficiently synchronize different cell types [32]. When placental macrophages were incubated with 50% FBS for 2 h before CT0, the circadian expression of the main clock gene, BMAL1, was restored (Appendix A), demonstrating the ability of the serum to resynchronize placental macrophages. 

As trophoblasts and macrophages interact within placental tissue [15], we wondered if co-culture of placental macrophages, devoid of circadian rhythm of clock genes, with trophoblasts may restore rhythmicity of clock genes in macrophages. Thus, we co-cultured macrophages with trophoblasts in autologous or heterologous conditions for 4 days and we next measured the expression of the clock genes. In 50% of the cases, we did not observe circadian rhythm from macrophages including *BMAL1*, *CLOCK,* and *NR1D1* genes with no differences in homologous and heterologous conditions (Figure 2C and Table 1). Whatever the conditions of coculture, some placental macrophages remained refractory to signals provided by trophoblasts but could be re-synchronized by serum shock. 

### 2.3. Circadian Rhythm Impairment in Preeclamptic Macrophages 

As circadian rhythms of different biological parameters are altered in preeclampsia [33], we hypothesized that placental macrophages isolated from women who developed preeclampsia presented an altered circadian rhythm of clock genes. We found absence of circadian rhythm of *BMAL1* in all tested placentas (*n* = 9) (Figure 3A). The amplitude was low (0.27) and the acrophase occurred at CT16 but was not statistically significant. The same result was observed for *PER2*, *CRY2,* and *NR1D1* clock genes (Figure 3A and Table 2). Finally, we reasoned that the co-culture of preeclamptic macrophages with trophoblasts from healthy placenta may restore the rhythmic oscillations of clock genes. However, this coculture did not restore the circadian rhythm of all the clock genes (Figure 3B and Table 2). These results demonstrated that preeclampsia is associated with alteration of the internal circadian clock in placental cells.

### 2.4. Melatonin Production by Placental Cells

We studied melatonin, a key hormone involved in placental homeostasis during pregnancy [28]. We report that trophoblasts and placental macrophages from healthy donors expressed the genes encoding enzymes (AANAT and ASMT) necessary for the production of melatonin at the transcriptomic level (Figure 4A). In placental macrophages from preeclamptic women, the *AANAT* enzyme gene was not expressed and *ASMT* expression was decreased compared to the healthy control. Moreover, placental cells from healthy donors strongly expressed the melatonin MT2 receptor which was decreased in preeclamptic placentas (Figure 4B), The MT1 receptor was not expressed. Finally, we measured the secretion of melatonin in supernatants of trophoblasts and macrophages. We found trophoblasts and macrophages from healthy donors produced melatonin, which exhibited circadian oscillations with a peak at CT15 (Figure 4C and Table 3). In contrast, no rhythmicity of melatonin production was observed in preeclamptic macrophages. It is noteworthy that healthy macrophage secreted significantly more melatonin than trophoblast cells as determined by mesor measurement (Figure 4D). However, the melatonin concentration was found to be strongly decreased in preeclamptic macrophage compared to a healthy donor. Taken together, these results showed that placental cells produced melatonin and expressed its MT2 receptor with a circadian rhythmicity. This circadian response was impaired in preeclampsia.

## 3. Discussion

In this study, we investigated the circadian rhythm of placental cells from human full-term placenta. In the literature, very few studies have been conducted on the rhythmicity of the placenta [21,34] and its involvement in the success of pregnancy. We showed here the circadian rhythm of clock genes in placental macrophages with a peak of expression during the day. This finding may be related to papers elsewhere published. Most of them have been conducted in animals in which the organization of placenta is markedly distinct from humans [35,36,37]. The pregnancy growth rate in the rat showed oscillations while more than two thirds of the daily weight gain of the fetus occurred during the active phase (at night) [38]. Balsalobre et al. showed the expression of the PER2 and DEC1 clock genes in a HTR-8/SVneo cell line derived from human trophoblasts [32]. Perez et al. explored the expression of CLOCK, BMAL1, PER2, and CRY1 clock genes in human placenta that they recovered at 00, 04, 08, 12, 16, and 20 h. They showed that they expressed the clock genes but only BMAL1 and CLOCK exhibited a significant circadian rhythm. The acrophase of the mRNA of BMAL1 and CLOCK rhythms appeared during the active phase (daytime) at 8 and 12 a.m., respectively [20]. Taken together these studies underline the difficulty of circadian rhythm analysis because of methodological similarities and differences. The use of different models from cell lines to animal models leads to divergent results regarding the oscillations and their synchronization in placental tissue. In addition the delivery time is a source of study heterogeneity as illustrated by S. Perez et al. [20].

In our study we reported rhythmic and non-rhythmic placental macrophages. This may be explained by the presence of subpopulations of macrophages in terms of origin (decidual macrophages, Hofbauer cells) or in terms of polarization. Indeed Lellupitiyage Don et al. reported that the circadian rhythm of macrophages is influenced by their polarization status [39] The authors reported that the M1 macrophage showed a significant loss of rhythmicity, while the M2 profile is associated with an increase in amplitude and period. The link between variations in macrophage polarization and their circadian rhythm can be explained by the presence of melatonin since it has recently been shown that this latter is secreted by microglia and drives the expression of pro-inflammatory mediators by macrophages [40]. Studies should be considered to address how circadian oscilliations might change macrophage phenotype and vice versa.

We showed that the placental macrophages no longer exhibited circadian rhythmicity after 4 days of culture. It is known that cells cultured under constant temperature conditions show a loss of circadian rhythm progressively [31]. On the contrary, trophoblasts retained the rhythm (data not showed), suggesting the presence of an intrinsic control independent of external stimuli or the secretion of a soluble molecule capable of synchronizing them. The co-culture of macrophages with trophoblasts restored the circadian rhythm in about 50% of placentas. Indeed, trophoblasts play a role in placental function and form the placental niche with macrophages and other cell types [41,42] but a large proportion of placental macrophages remained refractory to trophoblast signals. Although the serum shock also re-synchronizes macrophages and induces circadian expression of clock genes, the mechanisms involved in this synchronization are likely to be different and remain not understood. Moreover, the addition of trophoblast culture supernatant did not restore circadian rhythms of macrophages, demonstrating that it is the presence of cells that induces the rhythm. One of the candidates is melatonin. Indeed, it has been showed that trophoblasts (i.e., villous trophoblasts, cytotrophoblasts and syncytiotrophoblasts, and choriocarcinoma cell lines JEG-3 and BeWo) produce melatonin and express its receptors and functional melatonin enzymes for the synthesis enzymes AANAT and ASMT [27,29]. This local production may occur through paracrine and autocrine pathways [43]. Interestingly the enzyme activity was found higher in trophoblasts than in placenta tissue suggesting that these cells might be the principal source of placenta melatonin that may contribute to melatonin concentrations in maternal blood during pregnancy [27]. We showed that melatonin secretion by trophoblast exhibited circadian rhythm. This finding was in accordance with an increase in melatonin level in maternal blood in both the diurnal and nocturnal period [27,44,45]. In contrast, during pregnancy complications in the presence of a hypoxic environment, such as preeclampsia, melatonin production by trophoblasts was decreased [46]. The production of melatonin by macrophages in this context could be a compensatory alternative for the placental microenvironment. Further investigations are needed to clarify the role of melatonin secreted by trophoblasts and macrophage during normal and complicated pregnancy. 

Recently, Zhou et al. reported difference in gene expression between placenta with preeclampsia versus non-preeclampsia [47]. They reported decrease of *CLOCK*, *CRY1*, *NR1D2,* and *PER3* transcripts associated with altered hypoxia, autophagy, cell migration/invasion, and membrane trafficking pathways in preeclamptic placenta. Placental macrophages from women who developed preeclampsia during pregnancy did not show significant circadian rhythm of clock genes. Co-culture of these macrophages with trophoblasts from healthy women did not induce a circadian rhythm, suggesting a defect in the molecular clock of these cells. Studies have shown a disruption of the molecular clock during preeclampsia. The methylation of DNA of circadian clock and clock-controlled genes was altered in early preeclampsia compared to spontaneous preterm birth in placental tissue [48]. Early preeclampsia is also associated with altered epigenetic programming of the circadian clock pathway in umbilical, endothelial, and placental cells [48]. In addition, melatonin is a key hormone of the circadian rhythm and pregnancy. It is produced by the ovaries and in the placenta and protects against molecular mutations [28]. In preeclampsia, this hormone is known for its antioxidant capacity [49], however, further investigations are needed to reveal its involvement in synchronizing molecular clocks of placenta. A disturbance of the placental circadian rhythm could affect fetal development and may be linked to the development of obstetrical pathologies [50]. Thus, the circadian rhythm may be a witness of the cooperation between trophoblasts and macrophages. It may be a biomarker for preeclampsia.

It was proposed that maternal circadian rhythms entrain rhythms in the fetus whose candidates include glucocorticoids, dopamine, and melatonin [21]. The placenta transmits maternal rhythmic signals to the fetus [23]. Here, we reported circadian rhythms in placental cells and melatonin production by placental macrophages. Our finding supports the hypothesis that the placenta presents its own rhythmic function which could be regulated by melatonin secreted by placental macrophages. Interestingly, the lack of melatonin production from mouse strains or pinealectomy of rat did not impact on the daily rhythms in physiology and behavior which appeared synchronized among the offspring [51,52,53]. These findings suggested that other candidates must be assessed at the placental level.

During preeclampsia we report a disruption of the circadian rhythm of placental cells. Recently Zhou et al. illustrated the link between preeclampsia (hypertension) and clock genes expression [47]. It would be interesting to evaluate the link between preeclampsia and circadian rhythm by assessing the impact between patients developing early- or late-onset preeclampsia. Nevertheless, some limiting points should be emphasized in this study such as placenta sampling or the method to investigate the circadian rhythm which remains a problem to be solved in the studies on chronobiology. Taken together, this study highlights the links between circadian rhythmicity and pregnancy and the relevance of placenta-synthesized melatonin on healthy and complicated pregnancy such as preeclampsia.

## 4. Materials and Methods

### 4.1. Placenta Collection

Twenty healthy pregnant women and nine patients with preeclampsia were included in the study after receiving informed consent and the approval of the study by the “Comité d’Ethique d’Aix-Marseille Université” (number 08-012). Placentas were collected in the Gynecology-Obstetrics Department of the hospital Conception and the Saint-Joseph hospital (Marseille, France). The clinical characteristics of normotensive healthy pregnant women and preeclampsia patients are presented in Table 4. Healthy pregnant women did not present any significant facts during pregnancy. In contrast, the group of pre-eclampic patients had hypertension during pregnancy for 77.7% of them. No differences in terms of BMI, management, and parity were observed between the two groups. We observed significant differences for the weight at birth between the two groups (*p* = 0.0001).

### 4.2. Cell Isolation and Culture

Placental tissue on the maternal side was cut into small pieces, and digested in Hank’s balanced saline solution (HBSS, ThermoFisher, France) containing 2.5 mM DNase I and 2.5% trypsin (Life Technologies, Carlsbad, CA, USA) at 37 °C as previously described [54,55]. After filtration through 100 μm pores, total cells were recovered, and placental macrophages were isolated. Briefly, the cell suspension was deposited on Ficoll cushion (Eurobio, Les Ulis, France), centrifuged to recover mononuclear cells, and macrophages were isolated using magnetic beads coated with anti-CD14 antibodies Miltenyi Biotec, Bergisch Glabach, Germany) as previously described [15,56]. The purity of the isolated CD14+ placental macrophages was evaluated by flow cytometry and was greater than 98%.

For trophoblast isolation, placental cell suspension was deposited on a 25–60% Percoll cushion (GE Healthcare, Tremblay-en-France, France) as previously described [57]. Collected cells were enriched with an anti-human IgG2a+b directed against epidermal growth factor (EGFR) antibody and secondary antibody anti-mouse IgG2a+b magnetic microbeads (Miltenyi Biotec). Isolated macrophages and trophoblasts were frozen at −80 °C in 10% dimethylsulfoxide and 90% fetal bovine serum (FBS, Life Technologies) before experiments with 2 × 10^7^ cells per cryotube.

The blood samples (leukopacks) used in our study were provided by the French national blood bank, the Établissement Français du Sang (EFS), which carries out donor inclusions, ensures that informed consent is granted, and collects samples. Through an agreement established between our laboratory and the EFS (No. 7828), buffy coats were obtained from healthy volunteers. PBMCs were isolated on Ficoll (Eurobio, Les Ulis, France) cushions, frozen using 10% dimethyl sulfoxide (DMSO) and 90% FBS, and then stored at −80 °C. 

Monocytes were isolated from PBMCs as previously described [58]. Briefly, the PBMCs were thawed, rinsed, and then incubated in 24-well plates (1 × 10^6^ cells/well) at 37 °C in RPMI 1640 containing 10% FBS, 2 mM glutamine, 100 U/mL penicillin, and 50 µg/mL streptomycin (Life Technologies, USA). After 2 h, non-adherent cells were discarded, and adherent monocytes were washed. The purity of the monocytes was >95%, as revealed by the expression of the monocyte marker, CD14, by flow cytometry. Monocytes were also differentiated into macrophages following incubation in RPMI 1640 containing 10% inactivated human AB serum (MP Biomedicals, Solon, OH, USA), 2 mM glutamine and antibiotics for three days and in RPMI 1640 supplemented with 10% FBS for four more days, as previously described [59]. More than 95% of the differentiated cells were MDMs, as determined by the expression of CD68, a pan-macrophage marker [60].

### 4.3. Study Design of Circadian Rhythm in Placental Cells

Placental macrophages (5 × 10^5^ cells/well) were thawed at 12 a.m. and cultured in Dulbecco’s modified Eagles’s medium (DMEM) supplemented with 10% FBS, 100 IU/mL penicillin, and 50 µg/mL streptomycin (Life Technologies) at 37 °C. The experiment started at midnight 12 p.m. equivalent to Circadian Time 0 (CT0), and samples were analyzed for molecular clock genes every 4 h for 24 h. In some experiments, macrophages and trophoblasts were co-cultured using the transwell system (Sigma-Aldrich, Saint-Quentin-Fallavier, France) with 0.4 µm porosity according to the manufacturer’s recommendations. The cells were co-cultured 4 days at 37 °C and the experiments began at CT0 of day 5 as illustrated in the Appendix A.

### 4.4. Study of Circadian Gene Expression

Total RNA was extracted from lysed cells using the RNA Mini Kit (Qiagen, Courtaboeuf, France) and a DNase I treatment to eliminate DNA contaminants as previously described [61]. The quantity and quality of RNAs were evaluated using a Nanodrop spectrophotometer (Nanodrop Technologies, Wilmington, DE, USA). Complementary DNA was generated using a reverse transcription of RNAs with MMLV kits (Life Technologies). A quantitative real-time PCR (qRT-PCR) was performed using a Smart SYBRgreen kit (Roche Diagnostics, Meylan, France) and a CFX Touch Real-Time PCR Detection System (Bio-Rad, Gémenos, France) using specific primers listed in Appendix A as previously described [62]. The analyses were carried out using the standard curve method with ACTB gene encoding β-actin as the endogenous normalizing control. The results are expressed according to the following formula: fold change = 2^−ΔΔCt^; with Ct = cycle threshold, ΔCt = Ct_target_ − Ct_actin_ and ΔΔCt = ΔCt of each time point −ΔCt of the maximum value between the 6 time points.

### 4.5. Study of Melatonin and Its Receptors

After amplification of the MT2 melatonin receptor gene and the actin as positive control by real-time PCR, a migration was performed on a 2% agarose gel (Lonza, Morristown, NJ 07960, USA) on a Mupid One electrophoresis chamber. Melatonin levels were determined in the culture supernatants using an ELISA based on a competitive enzyme immunoassay technique according to the manufacturer’s instructions (Aviva Systems Biology, San Diego, CA 92121, USA). The sensitivity of the assay was 15.6 pg/mL.

### 4.6. Statistical Analysis

Analyses of clock gene rhythms were performed using R studio version 4.0 (v3.4.0, Boston, MA, USA). Cosinor transformation was used to estimate for a given variable its variations over a 24-h period [2]. Circadian rhythm parameters were investigated including the acrophase (time elapsed until a maximum activity) and its inverse the batyphase, the amplitude (half of the maximum variation of the considered rhythm), and the mesor (average gene expression). A significant circadian rhythm is defined when the three circadian parameters are statistically significant. Continuous variables were expressed as medians ± interquartile, and comparisons between two groups were made using the Mann–Whitney non-parametric test. Statistical significance was defined for a threshold *p* ≤ 0.05.

## Figures and Tables

**Figure 1 ijms-24-02363-f001:**
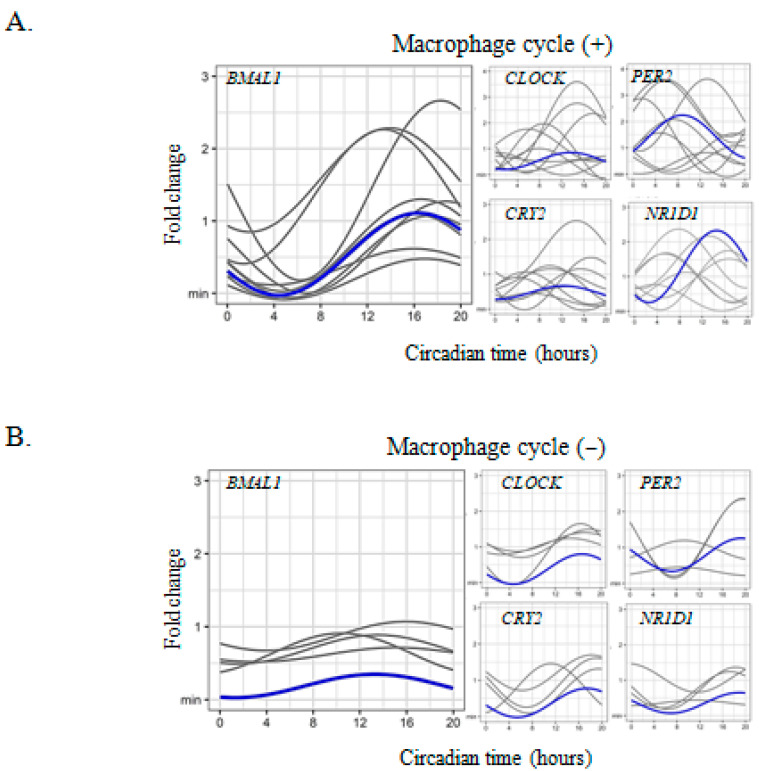
Circadian clock gene expression in placental macrophages. (**A**,**B**) macrophages (n = 13, including (**A**) *n* = 9 and (**B**) *n* = 4 for macrophages with or without cycle respectively) were isolated from placentas from healthy women and the expression of the clock genes (BMAL1, CLOCK, PER2, CRY2, and NR1D1) was evaluated with qRT-PCR every 4 h for 24 h and expressed in fold change from the maximum value. Each sinusoid (in gray) represents the expression of the clock genes over time for each donor after adjustment of the values by the cosinor model (mean of 3 technical replicates for each donor). The mean of the fold change of biological replicates was represented in blue. The statistical analysis was performed using the cosinor function using R studio (v3.4.0, Boston, MA, USA).

**Figure 2 ijms-24-02363-f002:**
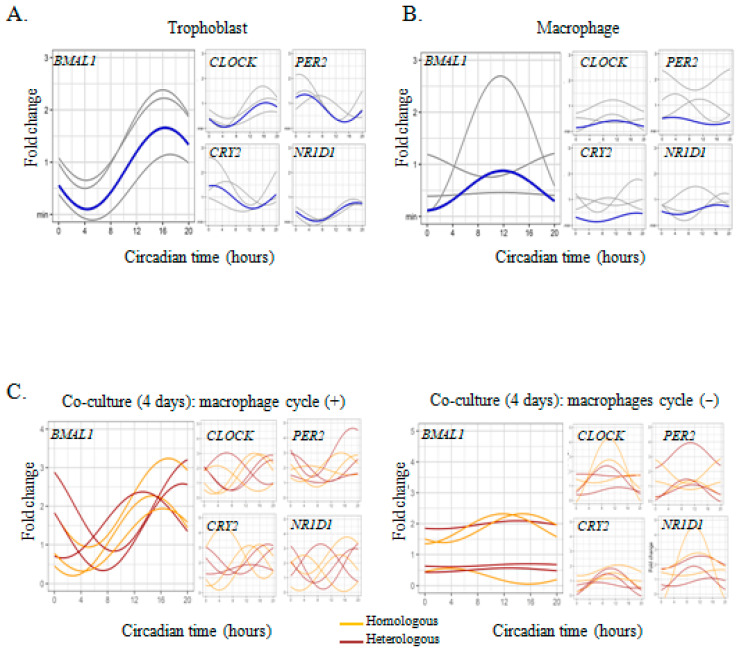
Circadian clock genes expression in macrophages co-cultured with trophoblasts. (**A**,**B**) Trophoblasts and macrophages were isolated from placentas from 3 healthy women and cultured for 4 days. Each sinusoid (in gray) represents the expression of the clock genes over time for each donor after adjustment of the values by the cosinor model (mean of 3 technical replicates for each donor). The mean of the fold change of biological replicates is represented in blue. (**C**) Macrophages (*n* = 6) were co-cultured with trophoblasts, in autologous (orange) or heterologous (brown) condition, for 4 days and the expression of clock genes was evaluated by qRT-PCR every 4 h for 24 h and expressed in fold change from the maximum value. Each sinusoid represents the expression of the clock genes over time for each donor after adjustment of the values by the cosinor model (mean of 3 technical replicates for each donor). The statistical analysis was performed using the cosinor function using R studio (v3.4.0, Boston, MA, USA).

**Figure 3 ijms-24-02363-f003:**
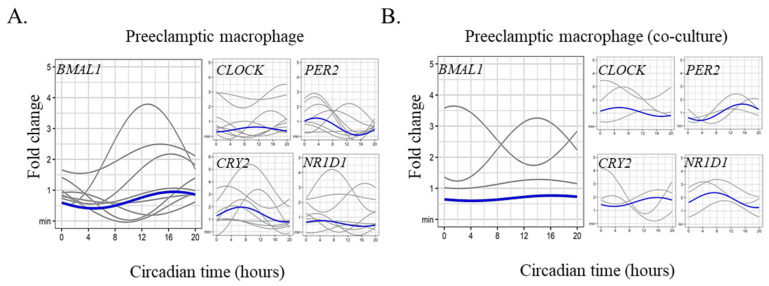
Preeclamptic placental macrophages and circadian clock gene expression. Placental macrophages were isolated from nine women who developed preeclampsia and (**A**) single culture (*n* = 9) or (**B**) co-culture with trophoblasts (*n* = 3, heterologous) were performed. The expression of the clock genes (BMAL1, CLOCK, PER2, CRY2, and NR1D1) was evaluated by qRT-PCR every 4 h for 24 h and expressed in fold change from the maximum value. Each sinusoid (in gray) represents the expression of the clock genes over time for each donor after adjustment of the values by the cosinor model (mean of 3 technical replicates for each donor). The mean of the fold change of biological replicates is represented in blue. The statistical analysis was performed using the cosinor function using R studio (v3.4.0, Boston, MA, USA).

**Figure 4 ijms-24-02363-f004:**
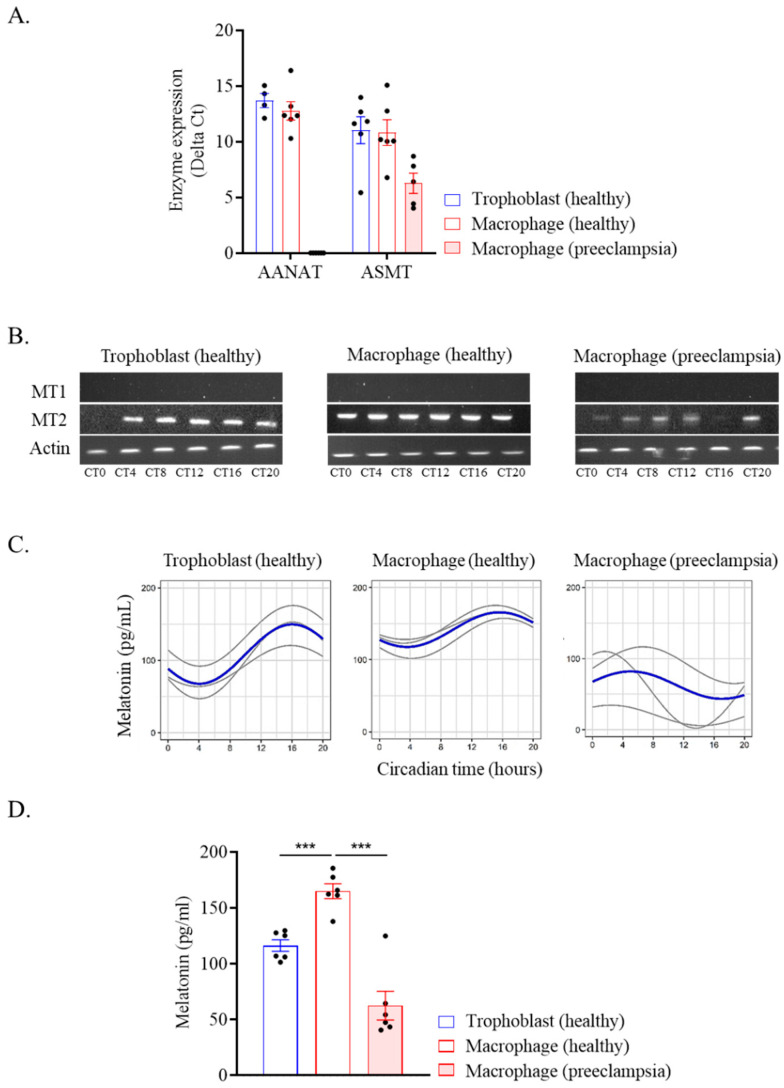
Melatonin from placental cells. Trophoblasts and macrophages were isolated from healthy and preeclamptic placentas. (**A**) The expression of AANAT and ASMT enzymes was evaluated by qRT-PCR every 4 h for 24 h and expressed in delta Ct (in technical triplicate for each donor and time). The average of enzymatic expression or each donor investigated was illustrated for trophoblast (*n* = 6), macrophage (*n* = 6), and preeclamptic macrophage (*n* = 5) (mean ± SEM). (**B**) The expression of the melatonin receptor MT1 and MT2 was evaluated by qRT-PCR every 4 h for 24 h and a migration was performed after gene amplification. (**C**,**D**) Melatonin level was quantified in the cell culture supernatant every 4 h for 24 h and expressed in pg/mL. (**C**) The circadian rhythmicity of the melatonin level was evaluated for 3 donors (with 3 technical replicates). Sinusoids in gray represent each investigated donor and the mean of biological replicates is represented in blue. The statistical analysis was performed using the cosinor function using R studio (v3.4.0, Boston, USA). (**D**) The mean of melatonin production over 24 h was evaluated for 6 donors (3 technical replicates mean ± SEM). *** *p* < 0.001.

**Table 1 ijms-24-02363-t001:** Rhythmic parameters (mesor, amplitude and acrophase) for placental macrophages. A *p* value < 0.05 for mesor, amplitude, and acrophase implies a rhythmicity of the investigated gene.

Conditions	Gene	Cycle	Mesor	CI 95%	*p*	Amplitude	CI 95%	*p*	Acrophase	CI 95%	*p*
**/**	** *BMAL1* **	+	0.80	(0.62; 0.98)	**<0.001**	0.56	(0.31; 0.83)	**<0.001**	16.38	(14.63; 18.14)	**<0.001**
		−	0.70	(0.53; 0.88)	**<0.001**	0.16	(−0.08; 0,41)	0.20	13.25	(7.39; 19.11)	0.67
	** *CLOCK* **	+	0.36	(0.16; 0.55)	**<0.001**	0.23	(−0.04; 0.51)	0.09	17.11	(12.56; 21.66)	**0.02**
		−	1.21	(0.66; 1.76)	**<0.001**	0.49	(−0.2 9; 1.27)	0.22	6.21	(0.10; 12.32)	0.06
	** *CRY2* **	+	0.73	(0.50; 0.96)	**<0.001**	0.75	(0.43; 1.07)	**<0.001**	17.27	(15.63; 18.87)	**<0.001**
		−	0.93	(0.10; 1.75)	**0.027**	0.27	(−0.90; 1.43)	0.65	9.379	(4.77; 13.99)	0.75
	** *PER2* **	+	6.70	(4.09; 9.31)	**<0.001**	2.74	(−0.95; 6.43)	0.14	8.46	(3.32; 13.60)	0.18
		−	1.14	(0.47; 1.81)	**<0.001**	0.26	(−0.68; 1.21	0.58	17.12	(15.38; 18.86)	0.46
	** *NR1D1* **	+	0.89	(0.77; 1.01)	**<0.001**	0.85	(0.68; 1.03)	**<0.001**	16.55	(15.77; 17.32)	**<0.001**
		−	0.87	(0.32; 1.42)	**0.002**	0.19	(−0.58; 0.97)	0.61	15.93	(13.00; 18.86)	0.60
**4 days of culture**	** *BMAL1* **	/	0.93	(0.51; 1.35)	**<0.001**	0.38	(−0.21; 0.97)	0.21	11.87	(5.88; 17.85)	0.96
	** *CLOCK* **		0.42	(0.07; 0.77)	**0.01**	0.03	(−0.46; 0.53)	0.89	11.69	/	0.99
	** *CRY2* **		0.87	(0.42; 1.33)	**<0.001**	0.36	(−0.27; 0.10)	0.26	11.02	(4.30; 17.73)	0.77
	** *PER2* **		0.40	(0.14; 0.66)	**0.003**	0.28	(−0.09; 0.65)	0.13	6.28	(1.28; 11.28)	**0.02**
	** *NR1D1* **		0.58	(0.15; 1.01)	**0.008**	0.22	(−0.39; 0.82)	0.48	21.43	/	0.63
**Co-culture with trophoblasts** **(4 days)**	** *BMAL1* **	+	1.58	(1.23; 1.92)	**<0.001**	0.8	(0.31; 1.30)	**0.001**	17.20	(14.87; 19.55)	**<0.001**
		−	1.21	(0.72; 1.71)	**<0.001**	0.14	(−0.56; 0.84)	0.70	12.99	(5.39; 20.60)	0.92
	** *CLOCK* **	+	1.70	(0.89; 2.50)	**<0.001**	1.18	(0.04; 2.33)	**0.04**	7.36	(3.68; 11.04)	**0.01**
		−	1.46	(−0.54; 3.45)	0.15	0.20	(−2.63; 3.03)	0.89	6.44	/	0.84
	** *CRY2* **	+	4.69	(2.47; 6.91)	**<0.001**	1.88	(−1.26; 5.03)	0.24	11.12	(4.73; 17.51)	0.79
		−	1.38	(−0.06; 2.82)	**0.06**	1.04	(−0.99; 3.07)	0.32	12.31	(4.83; 19.79)	0.93
	** *PER2* **	+	2.03	(1.28; 2.79)	**<0.001**	0.93	(−0.14; 1.99)	**0.09**	11.58	(7.18; 15.98)	0.85
		−	1.07	(−0.07; 2.21)	0.06	0.28	(−1.33; 1.89)	0.73	10.65	(0.72; 20.58)	0.90
	** *NR1D1* **	+	1.82	(1.06; 2.59)	**<0.001**	1.16	(0.07; 2.25)	**0.03**	6.99	(3.41; 10.57)	**0.006**
		−	1.48	(−0.42; 3.38)	0.12	0.27	(−2.42; 2.96)	0.12	17.57	/	0.77

+ = significant *BMAL1* cycle. − = No significant *BMAL1* cycle.

**Table 2 ijms-24-02363-t002:** Rhythmic parameters (mesor, amplitude, and acrophase) for preeclamptic macrophages. A *p* value < 0.05 for mesor, amplitude, and acrophase implies a rhythmicity of the investigated gene.

Gene	Cycle	Mesor	CI 95%	*p*	Amplitude	CI 95%	*p*	Acrophase	CI 95%	*p*
** *BMAL1* **	Single	1.10	(0.80; 1.40)	**<0.001**	0.27	(−0.15; 0.70)	0.20	16.85	(10.97; 22.73)	0.10
	Co-culture	2.03	(1.30; 2.75)	**<0.001**	0.08	(−0.95; 1.11)	0.87	16.18	/	0.10
** *CLOCK* **	Single	0.40	(0.25; 0.55)	**<0.001**	0.51	(0.30; 0.72)	**<0.001**	8.66	(7.07; 10.24)	**<0.001**
	Co-culture	0.79	(0.16; 1.42)	**0.014**	0.46	(−0.43; 1.35)	0.31	13.55	(6.16; 20.95)	0.68
** *CRY2* **	Single	0.77	(0.40; 1.14)	**<0.001**	0.28	(−0.25; 0.80)	0.29	14.41	(7.20; 21.63)	0.51
	Co-culture	4.12	(1.22; 7.03)	**0.005**	2.34	(−1.77; 6.44)	0.26	11.63	(4.92; 18.35)	0.91
** *PER2* **	Single	0.52	(0.25; 0.79)	**<0.001**	0.44	(0.05; 0.83)	**0.02**	12.53	(9.17; 15.88	0.75
	Co-culture	1.31	(0.58; 2.03)	**<0.001**	0.99	(−0.04; 2.02	**0.06**	14.91	(10.95; 18.88)	0.14
** *NR1D1* **	Single	0.53	(0.26; 0.81)	**<0.001**	0.36	(−0.03; 0.75)	**0.07**	13.53	(9.37; 17.69)	0.46
	Co-culture	1.10	(0.42; 1.79)	**0.002**	0.66	(−0.31; 1.63)	0.18	10.07	(4.49; 15.65)	0.49

**Table 3 ijms-24-02363-t003:** Rhythmic parameters (mesor, amplitude, and acrophase) of melatonin (pg/mL) circadian expression. A *p* value < 0.05 for mesor, amplitude, and acrophase implies a rhythmicity of the investigated gene.

Cells	Mesor	CI 95%	*p*	Amplitude	CI 95%	*p*	Acrophase	CI 95%	*p*
**Trophoblast (healthy)**	113.35	(96.92; 129.78)	**<0.001**	25.2816	(2.04; 48.52)	**0.03**	18.28	(14.77; 21.79)	**0.001**
**Macrophage** **(healthy)**	141.74	(130.12; 153.36)	**<0.001**	23.9745	(7.54; 40.41)	**0.004**	15.58	(12.96; 18.20)	**0.007**
**Macrophage** **(preeclampsia)**	48.32	(26.82; 69.82)	/	24.0605	(−6.34; 54.46)	0.12	2.99	/	0.22

**Table 4 ijms-24-02363-t004:** Clinical characteristics of patients with preeclampsia and control population.

	Control (*n* = 20)	Preeclampsia (*n* = 9)
Maternal age (year)	32 [24–43]	33 [24–44]
Gestational age (week)	39 [36–41]	36 [34–41]
BMI	25.75 [17.3–37]	28.75 [20.3–43.3]
Cesarean (*n*)	0	4
Weight at birth (g)	3430 [2740–4160]	2410 [1640–3260]

The maternal age and gestational age are given in median ± interquartile.

## Data Availability

Not applicable.

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
