# Peer review of "Disruption of the Expression of the Placental Clock and Melatonin Genes in Preeclampsia"

_ijms, 2023, doi:10.3390/ijms24032363_

Round 1

Reviewer 1 Report

Chrono-adjustment of the placenta in health and disease is a highly relevant topic. As mentioned by the authors not many scientific studies focused on placental clock genes and melatonin production. The title of the manuscript suggests that melatonin plays a key role in the disruption of the placental clock in preeclampsia. However, there is no data that support any relationship between clock genes and melatonergic biosynthetic enzymes coding genes. Lately, there is a suggestion to explore the present hypothesis.

One of the most interesting points of this manuscript is that clock genes and genes that codify biosynthetic melatonin enzymes by macrophages in culture exhibit a circadian rhythm. The authors were aware of the relevance of the data, however, they just emphasized it by reinforcing the obvious. Reading the discussion it is clear that there are many points that could be explored. Instead of just pointing out that " This finding may be related to papers elsewhere published.; page 11, line 332 " it will be highly relevant to clearly show the similarities and differences. Another point that could be explored is the work of Perez et al. (20). They established the rhythm using placenta obtained at different hours of the day, and the present manuscript shows autonomous oscillators in macrophages and trophoblasts. The clock genes expressed are different, and trophoblasts may drive macrophage expression. As the discussion follows the authors propose that the rhythm of the placenta is autonomous and not linked to the circadian timing of the whole body driven by neural (suprachiasmatic nuclei) and humoral (pineal gland, melatonin; adrenal, cortisol). I suggest to remove phrases that valorize the aim of the present paper to the detriment of others ("However, their study is limited by the fact that the authors used several placentas delivered at different times to explore the rhythm. Additionnaly, their observations of cycling gene in placenta tissue did not establish that there was an autonomous oscillators suggesting that their observations could be the result of cyclic humoral or neuronal signaling emanating from the SCN. Here, we focused on primary placental cell that constitute a better way to examine autonomous circadian clocks from placenta tissue." Page 11 Lines 330-335). The authors should discuss the possibility of a local or even double (local and central) circadian control. An intriguing possibility is that in conditions where the mother loses the rhythm of the central oscillator and/or melatonin production by the pineal the placental macrophages may take over this role. Recently Zhou et al., (https://doi.org/10.1038/s41598-022-22507-3) emphasized the linkage between hypertension and the master clock.

Another point that should be explored and even speculated in the discussion is that some nonrhythmic macrophages could be induced to rhythmically express clock genes when co-cultured with trophoblasts from rhythmic women. Besides reinforcing the integration between macrophages and trophoblasts, it is suggested to point out that different subtypes of macrophages were described in the placenta. known to be present in the placenta and maintain their phenotype after so many days in culture. Recently, a paper cited by the authors (Yao et al., 2019 - citation 12) described that the proportion between M1 and M2 macrophages is responsible for sustaining feto-maternal immune tolerance.

In 2018, Sagrillo-Fagundes et al. (https://doi.org/10.1111/jpi.12520 ) published a paper entitled  "Melatonin modulated autophagy and inflammation protecting human placenta trophoblast from hypoxia/reoxygenation". As is known, pre-eclampsia is a syndrome associated with placenta hypoxia/reoxygenation. Trophoblasts isolated from normal placenta exposed to H/R in vitro reduced melatonin synthesis when compared to cells maintained in normoxia. In addition, the production of TNF, IL-6, and NFkB was also increased. Melatonin, as well as, NFkB inhibition increased autophagy. This paper concludes, "This study suggests that H/R, which is present in pregnancy complications, inhibits endogenous melatonin production, thereby contributing to reduced syncytiotrophoblast viability."

Taking together the several mechanisms for inducing the synthesis of melatonin in macrophages, it is mandatory to discuss the possibility that in the placenta, as in several other structures, a danger signal such as hypoxia/ reoxygenation could lead to a local synthesis of melatonin which is not necessarily rhythmic.

The greatest concern with the present manuscript is that the title did not fit with the data! The authors did not link melatonin findings with clock genes, thus, we could suppose those clock genes or their protein could act on melatonergic enzymes and vice-versa. As a matter of fact, here only the expression of the genes involved in melatonin synthesis is shown. This is not sufficient to disclose the relevance of melatonergic function. The activity or even the amount of the enzyme that converts serotonin into N-acetylserotonin (arylalkylamine N-acetyltransferase, AA-NAT, EC 2.3.1.87) does not always correlate with gene expression. I will mention only one of the first papers of Stehle, but many others were published in the last 16 years (Ackermann and Stehle, 2006, PMID: 16687310).

M2 Macrophages/ microglia are known to synthesize melatonin. Recently, it was shown that the conversion of M1 into M2 cultured microglia challenged with the pathogen-associated molecular pattern, lipopolysaccharide is driven by melatonin synthesized by the microglia, which act in its own receptor, reducing the expression of pro-inflammatory mediators (Souza et al., 2022 - doi: 10.32794/mr112500120).

The melatonin receptors evaluated in the present study are the G-protein coupled receptors. The protein codified by the genes evaluated acts as dimers. Therefore, it is not possible to extrapolate the function of the receptors by evaluating gene expression. Inasmuch, as observed with many other GPCRs the activity of the receptor is not necessarily sustained by gene expression. However, the data shown are interesting and deserve to be published. I will strongly suggest a revision of the text in order to include the relevance of nonrhythmic synthesized melatonin by macrophages and trophoblasts in pathophysiological conditions, besides elaborating on the role of endogenous placenta rhythm driven by macrophages and trophoblasts.

The data did not show the key role of melatonin, indeed, they show the "expression" of the genes of the enzymes that convert serotonin into N-acetylserotonin (AA-NAT) and melatonin (ASMT), as well as the genes that codify for MT1 and MT2 receptors. Regarding the enzyme AA-NAT, it is well-documented that the protein is directed to the proteasome as soon as it is synthesized. Thus transcription of the gene does not result in a synthesis increase. The protein need to be phosphorylated to be active. 

In conclusion, the results are interesting, and deserve to be published, because adds information to the area. 

Author Response

Comments and Suggestions for Authors

Chrono-adjustment of the placenta in health and disease is a highly relevant topic. As mentioned by the authors not many scientific studies focused on placental clock genes and melatonin production. The title of the manuscript suggests that melatonin plays a key role in the disruption of the placental clock in preeclampsia. However, there is no data that support any relationship between clock genes and melatonergic biosynthetic enzymes coding genes. Lately, there is a suggestion to explore the present hypothesis.

  1. One of the most interesting points of this manuscript is that clock genes and genes that codify biosynthetic melatonin enzymes by macrophages in culture exhibit a circadian rhythm. The authors were aware of the relevance of the data, however, they just emphasized it by reinforcing the obvious. Reading the discussion it is clear that there are many points that could be explored. Instead of just pointing out that " This finding may be related to papers elsewhere published: page 11, line 332" it will be highly relevant to clearly show the similarities and differences.

We thank the reviewer for this comment. As suggested we reported the differences and similarities between studies cited in the text in the discussion part of our revised manuscript.

  1. Another point that could be explored is the work of Perez et al. (20). They established the rhythm using placenta obtained at different hours of the day, and the present manuscript shows autonomous oscillators in macrophages and trophoblasts. The clock genes expressed are different, and trophoblasts may drive macrophage expression. As the discussion follows the authors propose that the rhythm of the placenta is autonomous and not linked to the circadian timing of the whole body driven by neural (suprachiasmatic nuclei) and humoral (pineal gland, melatonin; adrenal, cortisol). I suggest to remove phrases that valorize the aim of the present paper to the detriment of others ("However, their study is limited by the fact that the authors used several placentas delivered at different times to explore the rhythm. Additionally, their observations of cycling gene in placenta tissue did not establish that there was an autonomous oscillators suggesting that their observations could be the result of cyclic humoral or neuronal signaling emanating from the SCN. Here, we focused on primary placental cell that constitute a better way to examine autonomous circadian clocks from placenta tissue." Page 11 Lines 330-335). The authors should discuss the possibility of a local or even double (local and central) circadian control. An intriguing possibility is that in conditions where the mother loses the rhythm of the central oscillator and/or melatonin production by the pineal the placental macrophages may take over this role. Recently Zhou et al., (https://doi.org/10.1038/s41598-022-22507-3) emphasized the linkage between hypertension and the master clock. M2 Macrophages/ microglia are known to synthesize melatonin. Recently, it was shown that the conversion of M1 into M2 cultured microglia challenged with the pathogen-associated molecular pattern, lipopolysaccharide is driven by melatonin synthesized by the microglia, which act in its own receptor, reducing the expression of pro-inflammatory mediators (Souza et al., 2022 - doi: 10.32794/mr112500120).

As suggested by the reviewer we removed the paragraph about the work of Peres et al. We also discussed the possibility of a local or even double (local and central) circadian control. We also thank the reviewer for this interesting study from Zhou et al.

  1. Another point that should be explored and even speculated in the discussion is that some nonrhythmic macrophages could be induced to rhythmically express clock genes when co-cultured with trophoblasts from rhythmic women. Besides reinforcing the integration between macrophages and trophoblasts, it is suggested to point out that different subtypes of macrophages were described in the placenta known to be present in the placenta and maintain their phenotype after so many days in culture. Recently, a paper cited by the authors (Yao et al., 2019 - citation 12) described that the proportion between M1 and M2 macrophages is responsible for sustaining feto-maternal immune tolerance.

We thank the reviewer for this comment. Indeed, the different populations of macrophages could explain the presence of rhythmic and nonrhythmic macrophages. We discussed this point in the revised version of the manuscript.

  1. In 2018, Sagrillo-Fagundes et al. (https://doi.org/10.1111/jpi.12520) published a paper entitled "Melatonin modulated autophagy and inflammation protecting human placenta trophoblast from hypoxia/reoxygenation". As is known, pre-eclampsia is a syndrome associated with placenta hypoxia/reoxygenation. Trophoblasts isolated from normal placenta exposed to H/R in vitro reduced melatonin synthesis when compared to cells maintained in normoxia. In addition, the production of TNF, IL-6, and NFkB was also increased. Melatonin, as well as, NFkB inhibition increased autophagy. This paper concludes, "This study suggests that H/R, which is present in pregnancy complications, inhibits endogenous melatonin production, thereby contributing to reduced syncytiotrophoblast viability."Taking together the several mechanisms for inducing the synthesis of melatonin in macrophages, it is mandatory to discuss the possibility that in the placenta, as in several other structures, a danger signal such as hypoxia/ reoxygenation could lead to a local synthesis of melatonin which is not necessarily rhythmic.

We thank the reviewer for this comment that we included in the revised version of our manuscript

  1. The greatest concern with the present manuscript is that the title did not fit with the data! The authors did not link melatonin findings with clock genes, thus, we could suppose those clock genes or their protein could act on melatonergic enzymes and vice-versa. As a matter of fact, here only the expression of the genes involved in melatonin synthesis is shown. This is not sufficient to disclose the relevance of melatonergic function. The activity or even the amount of the enzyme that converts serotonin into N-acetylserotonin (arylalkylamine N-acetyltransferase, AA-NAT, EC 2.3.1.87) does not always correlate with gene expression. I will mention only one of the first papers of Stehle, but many others were published in the last 16 years (Ackermann and Stehle, 2006, PMID: 16687310).

We agree with the reviewer. We are proposing a new title: “Disruption of the expression of the placental clock and melatonin genes in preeclampsia”.

  1. The melatonin receptors evaluated in the present study are the G-protein coupled receptors. The protein codified by the genes evaluated acts as dimers. Therefore, it is not possible to extrapolate the function of the receptors by evaluating gene expression. Inasmuch, as observed with many other GPCRs the activity of the receptor is not necessarily sustained by gene expression. However, the data shown are interesting and deserve to be published. I will strongly suggest a revision of the text in order to include the relevance of nonrhythmic synthesized melatonin by macrophages and trophoblasts in pathophysiological conditions, besides elaborating on the role of endogenous placenta rhythm driven by macrophages and trophoblasts. After rereading your comment we think that the problem in this paragraph is the link between melatonin and rhythm that only appears at the end of paragraph. So we reintroduced that from the first sentence.

  1. The data did not show the key role of melatonin, indeed, they show the "expression" of the genes of the enzymes that convert serotonin into N-acetylserotonin (AA-NAT) and melatonin (ASMT), as well as the genes that codify for MT1 and MT2 receptors. Regarding the enzyme AA-NAT, it is well-documented that the protein is directed to the proteasome as soon as it is synthesized. Thus transcription of the gene does not result in a synthesis increase. The protein need to be phosphorylated to be active. 

We agree with the reviewer. We have modified some sentence in the revised version of the manuscript. Nevertheless, we also investigated the melatonin secreted in the culture supernatant at different time point in order to evaluate the circadian rhythm of this protein.

  1. In conclusion, the results are interesting, and deserve to be published, because adds information to the area.

Thank you for your reviewing

Reviewer 2 Report

This paper investigates Circadian rhythms in placenta in normal and preeclamptic pregnancy. It makes important observations that preeclampsia affects circadian oscillations of placental macrophages (clock genes) and modifies the production of melatonin. These novel and important findings could provide new insights into the pathophysiology of preeclampsia. My comments are as follows:

1.     Main limitation of this study is a small sample containing only 9 preeclampsia patients. Are there any plans to further validate these findings in future studies?

2.     A more detailed description of the cohort should be included: How did the patients differ in known risks of preeclampsia such as BMI, preexisting hypertension, race, gravida, whether it is a multiple gestation, etc.  Were all controls normotensive pregnancies? These should be added to Table 1. If there are differences, can these in any way impact the findings?

3.     Did these preeclamptic pregnancies include both early- and late-onset preeclampsia? If yes, was there any difference observed among those two groups?

4.     It is stated that placentas were collected at term (Section 2.1), but Table 1 indicates that some of preeclamptic pregnancies were preterm (GA<37 weeks). Please specify.

5.     I wonder if more general/agnostic approach looking at multiple cell types present in placenta and multiple placental hormones possibly using machine learning methods could provide a more comprehensive understanding. The authors should comment on this.

6.     Please provide further explanation of Table 2 and comment on non-significant p-values present also for BMAL1, CLOCK, CRY2 and NR1D1.

7.     It is unclear to me why figure captioning contain placenta numbers, for example “placentas N° 2-10”. Where is this information needed/used?

8.     Figure 4 is very telling.

The paper is very to the point, and the results and contribution to the field are clearly explained. I recommend acceptance.

Author Response

Reviewer 2

Comments and Suggestions for Authors

This paper investigates Circadian rhythms in placenta in normal and preeclamptic pregnancy. It makes important observations that preeclampsia affects circadian oscillations of placental macrophages (clock genes) and modifies the production of melatonin. These novel and important findings could provide new insights into the pathophysiology of preeclampsia. My comments are as follows:

  1. Main limitation of this study is a small sample containing only 9 preeclampsia patients. Are there any plans to further validate these findings in future studies?

It would be interesting to continue this study with a larger sample. Many avenues will be explored in the future including new ethical procedures. This limiting point was added in the conclusion of the revised version of our manuscript.

  1. A more detailed description of the cohort should be included: How did the patients differ in known risks of preeclampsia such as BMI, preexisting hypertension, race, gravida, whether it is a multiple gestation, etc. Were all controls normotensive pregnancies? These should be added to Table 1. If there are differences, can these in any way impact the findings?

As suggested by the reviewer we added clinical characterization of the cohort in the material and method part including the table 1 of our revised version of the manuscript.

  1. Did these preeclamptic pregnancies include both early- and late-onset preeclampsia? If yes, was there any difference observed among those two groups?

In our cohort, only one patient had early-preeclampsia, so we cannot assess the impact on our observations. However, we have included this in the discussion.

  1. It is stated that placentas were collected at term (Section 2.1), but Table 1 indicates that some of preeclamptic pregnancies were preterm (GA<37 weeks). Please specify.

Indeed, we have included placentas after term deliverance for the control group and variable for the preeclamptic group. We therefore remove the term " At term" from the revised manuscript.

  1. I wonder if more general/agnostic approach looking at multiple cell types present in placenta and multiple placental hormones possibly using machine learning methods could provide a more comprehensive understanding. The authors should comment on this.

We thank the reviewers for this suggestion and we agree that a more general exploration would allow a better understanding. Indeed, besides macrophages and trophoblasts, several other immune cells are present in the placenta and their rhythm is modulated by several hormones, including cortisol. Machine learning would allow to better understanding the mechanism of these rhythms as well as their interactions and their implication in placental pathologies.

  1. Please provide further explanation of Table 2 and comment on non-significant p-values present also for BMAL1, CLOCK, CRY2 and NR1D1.

We thank the reviewers for this suggestion. Table 2 represents the statistical results of the different placental macrophage rhythm parameters (me2sor, acrophase and amplitude). The cycle column indicates the presence of significant rhythm, it is + when the p value is significant for all 3 parameters for the Bmal1 gene. The CI column represents the confidence interval.

When the p value is not significant for Bmal1, it is considered that there is no significant rhythm for the placenta in question. On the contrary, when p is significant for Bmal1 on all 3 parameters, we deduce the presence of a significant rhythm even if the other genes are not significant since Bmal1 drives ClOCK and both CRY2 and NR1D1 may be negative factors that inhibit their own activation and therefore their rhythm.

  1. It is unclear to me why figure captioning contains placenta numbers, for example “placentas N° 2-10”. Where is this information needed/used?

We thank the reviewer for this comment. We deleted this information in the revised version of the manuscript.

  1. Figure 4 is very telling.

Thank you for this comment.

The paper is very to the point, and the results and contribution to the field are clearly explained. I recommend acceptance.

Round 2

Reviewer 1 Report

The revised manuscript has improved many of the issues raised. The data are interesting and its publication will support interconnections between clock genes, melatonin synthesis and pregnancy.

There are still two minor points that it would be important to correct. In the introduction, the two enzymes of melatonin synthesis are defined by names used, without synonymy or classification. The work does not require in-depth study of the subject.

The consistency of terminology must be maintained throughout the text. In line 74 the enzyme which transforms N-acetylserotonin into melatonin (ASMT) is defined properly. But in line 364 the authors use only the abbreviation of the synonym (HIOMT). I suppose that the present text follows the terminology used in the citation. However, there is no mention about this abbreviation and no mention that AMS and HIOMT are, indeed, the same enzyme. I suggest maintaining the term ASMT in line 364.

Finally, the last sentence could have a more accurate tone.  Instead of "thids study sheds a little more light on the links between.."

This study highlights the links between circadian rhythmicity and pregnancy and the relevance of placenta-synthesized melatonin on healthy and.....

Author Response

The revised manuscript has improved many of the issues raised. The data are interesting and its publication will support interconnections between clock genes, melatonin synthesis and pregnancy.

There are still two minor points that it would be important to correct. In the introduction, the two enzymes of melatonin synthesis are defined by names used, without synonymy or classification. The work does not require in-depth study of the subject.

The consistency of terminology must be maintained throughout the text. In line 74 the enzyme which transforms N-acetylserotonin into melatonin (ASMT) is defined properly. But in line 364 the authors use only the abbreviation of the synonym (HIOMT). I suppose that the present text follows the terminology used in the citation. However, there is no mention about this abbreviation and no mention that AMS and HIOMT are, indeed, the same enzyme. I suggest maintaining the term ASMT in line 364.

Finally, the last sentence could have a more accurate tone.  Instead of "thids study sheds a little more light on the links between.."

This study highlights the links between circadian rhythmicity and pregnancy and the relevance of placenta-synthesized melatonin on healthy and.....

 We thank the reviewer for these two comments. We have corrected them in the revised version of our manuscript. Thank you for this quality expertise.